



# Altered sub-seasonal predictability of Community Atmosphere Model 5 (CAM5) in CESM 1.2.1 by the choices of dynamical core

Ha-Rim Kim[1], Baek-Min Kim[2,*], Sang-Yoon Jun[3], Yong-Sang Choi[1,*]

[1]Department of Climate and Energy Systems Engineering, Ewha Womans University, Seoul, 03760, Republic of Korea
[2]Department of Environmental Atmospheric Sciences, Pukyong National University, Busan, 48513, Republic of Korea
[3]Unit of Arctic Sea-Ice Prediction, Korea Polar Research Institute, Incheon, 21990, Republic of Korea
[*]These authors contributed equally to this study.

*Correspondence to*: Baek-Min Kim (baekmin@pknu.ac.kr)

**Abstract.** This study investigates the prediction skill of sub-seasonal prediction models that vary based on the choice of two dynamical cores: the finite volume (FV) dynamical core on a latitude-longitude grid system and the spectral element (SE) dynamical core on a cubed-sphere grid system. Recent research showed that the SE dynamical core on a uniform grid system increases parallel scalability and removes the need for polar filters for mitigating uncertainty in climate prediction, particularly for the Arctic region. However, it still remains questionable whether the choice of dynamical cores can actually yield significant changes in prediction skill. To tackle this issue, we implemented a sub-seasonal prediction model based on the Community Atmospheric Model version 5 by incorporating the above two dynamical cores with virtually the same physics schemes. Sub-seasonal prediction skills of the SE dynamical core and FV dynamical core are verified with ERA-Interim reanalysis during the early winter (November–December) and the late winter (January–February) from 2001/2002 to 2017/2018. The prediction skills of two different dynamical cores were significantly different regardless of the similar physics scheme. In the ocean, the predictability of the SE dynamical core is similar to that of the FV dynamical core, mostly because our simulation configuration imposes the same boundary and initial conditions at the surface. Notable differences in the one-month predictability between the two cores are observed for the wintertime Arctic and mid-latitudes, particularly over North America and Eurasia continents. With a one-month lead, the SE dynamical core exhibited higher predictability over North America in late winter (r ≈ 0.45 in SE, r ≈ 0.10 in FV) whereas the FV dynamical core showed relatively higher predictability in East Asia and Eurasia in early winter (r ≈ 0.15 in SE, r ≈ 0.43 in FV). Therefore, we conclude that caution is needed when selecting the dynamical cores of sub-seasonal prediction models. Partially, these differences can be ascribed to the different manifestations of Arctic-mid-latitude linkage in the two dynamical cores; the SE dynamical core captures warmer Arctic and colder mid-latitudes relatively better than the FV dynamical core.



# 1 Introduction

Over the past decade, there has been a significant effort to predict the state of the atmosphere at the sub-seasonal timescales (2–4 weeks to 2 months) (Brunet et al. 2010; Kirtman et al. 2014; Vitart et al. 2017). This timescale fills the gap between medium-range weather and seasonal forecasts at both global and regional scales (Doblas-Reyes et al. 2013; Tian et al. 2017; Robertson and Vitart 2018; Bushuk et al. 2019). It also provides a valuable opportunity to inform decision-makers of, for example, any changes in the risks of extreme events, which can help optimize resource management decisions (Kim et al. 2012; Mariotti et al. 2018; Smith et al. 2019). Recently, many climate prediction model communities developed and provided significant improvements in the predictability of sub-seasonal time scales. Many international climate model institutes were started to compare coupled climate projections, providing numerous applications for modeling communities including the sub-seasonal to seasonal (S2S) project from the European Center for Medium-Range Forecast (ECMWF) (https://www.ecmwf.int/en/research/projects/s2s); the North American Multi-Model Ensemble (NMME) from the International Research Institute for Climate and Society (IRI) (https://www.cpc.ncep.noaa.gov/products/NMME/); and the Earth System Grid (ESG) at the National Center for Atmospheric Research (NCAR) (https://www.earthsystemgrid.org/). The Sub-Seasonal Experiment (SubX) data sets are accessible through a public archive at Columbia University's IRI Data Library (http://cola.gmu.edu/kpegion/subx/).

A top priority in forecasting is estimating and possibly reducing uncertainty (Leutbecher and Palmer 2008; Berner et al. 2011; Knutti 2018). Uncertainty is inevitable since all climate prediction models are based on physical principles and are generated with various assumptions (Sun et al. 2018). Notably, uncertainty is introduced in the construction of current state-of-the-art modeling systems from the lack of observations in the polar regions, leading to a limited understanding of the physical processes and their inevitably incorrect parameterization. Furthermore, recent studies show that both internal and forced atmospheric variabilities of Arctic weather are tightly linked to those of mid-latitudes, invoking large fluctuations in the jet-stream (Inoue et al. 2012; Kim et al. 2014; Mori et al. 2014; Sato et al. 2017; Jung et al. 2017). Several ongoing debates on the linkage issue, such as 'warm Arctic-cold continents (WACC)', is a good example of the considerable uncertainty that persists among models relevant to the Arctic climate system (Honda et al. 2009; Liu et al. 2012; Sun et al. 2015; Chen et al. 2018).

Among the issues of uncertainty in the Arctic climate system, recent studies suggest the choice of a dynamical core to improve the forecasting quality in the Arctic regions (Jun et al. 2018). The dynamical core resolves the fluid motion governing atmospheric dynamics on numerical equations based on the features of a grid structure (Harris and Lin 2013). Thus, there is increasing attention in highly scaling performance of dynamical cores with less structured or unstructured grids based on the uniform resolutions (Lin and Rood 1997; Lin 2004; Donner et al. 2011). For example, the use of the cubed-sphere grid in the dynamical core considerably enhances the computational efficiency, resolving pole singularity issues, compared to the latitude-longitude grid formation.



Jun et al. (2018) raised the issue of assessing the influence of existing dynamical cores underlying the grid formation of the Arctic region in a global simulation compared to the various global climate simulations. The use of different dynamical cores significantly affects the simulations of Arctic winter climate and linked teleconnection in the mid-latitudes. Notably, the spectral element core on a cubed-sphere grid simulates a warmer Arctic winter surface and a robust cooling response over
North America, unseen with a finite volume core on a latitude-longitude grid formation. These modeling results for the Arctic and mid-latitudes advise the need for more precise climate modeling and forecasting for the Arctic region.

This earlier suggestion has motivated the choice to model sub-seasonal predictions using different dynamical cores. We are interested in quantifying the possible changes in sub-seasonal predictability arising from the use of different dynamical cores of an atmospheric model. In this study, we contrasted results from two different dynamical cores with different grid
formations: a spectral element dynamical core on a cubed-sphere grid system and a finite volume core on a latitude-longitude grid system for feasible analysis.

This paper is structured as follows. We begin with describing the sub-seasonal prediction model design and analyses in Section 2. Section 3 presents the various results, focusing on comparing the predictability of the prediction model performance using the two dynamical cores. In Section 4, we discuss our findings and summarize this study. We also provide
questions for future work.

30



## 2 Model Description and Data

### 2.1 Model: Experimental Setup

#### 2.1.1 Dynamical cores

We strategically prepared two near-identical sub-seasonal prediction models. The only difference between them is

5 the choice of the dynamical core with the grid formation in the atmospheric model. The first uses the spectral element (SE) dynamical core on a cubed-sphere grid system with a horizontal resolution of 16 by 16 elements on one face and four collocation points on one element edge (e.g., named 'ne16np4', approximately 2º at the equator). The other uses the finite volume (FV) dynamical core on a latitude-longitude grid system with 91-latitudinal and 144-longitudinal grid points (e.g., named 'fv19') (Jun et al. 2018). The FV dynamical core is the present defaulting dynamical core in the Community Earth

System Model (CESM) and is being used for the CESM's contributions to the Intergovernmental Panel on Climate Change (IPCC) 5th assessment report (Collins et al., 2004; Denis et al., 2012). The SE dynamical core provides several profits compared to the FV one. As with all methods on cubed-sphere grid formation (quasi-uniform), preventing the fundamentally load-imbalanced polar filter permits for fixed and efficient two-dimensional formation disintegration, significantly improving a parallel scalability performance on calculations (Kay et al. 2016; Lauritzen et al. 2018). We implemented a sub-seasonal

prediction model by incorporating the two aforementioned dynamical cores with the same physics. The Community Atmosphere Model version 5 (CAM5) with CAM4 physics, which is an atmospheric model component of the CESM version 1.2.1, was used to perform sub-seasonal predictions.

#### 2.1.2. Topography

As for mapping the topography on different dynamical cores, technically, different dynamical cores need a different level of smoothing for the elevation data. In the FV dynamical core, the highest wavenumbers are removed by mapping to a latitude-longitude grid, whereas, in the SE dynamical core, the surface geopotential is smoothed by multiple applications (e.g., Laplace operator combined with a bound-preserving limiter, optimization-based mesh-improvement methods). Specifically, the mapping method applied to a SE dynamical core uses a strong high-order interpolation technique, which improves the

quality of the topography while retaining the integrity of the original surface approximation, thus, making a smoother topography than the FV dynamical core (Dennis et al. 2012; Choi and Hong. 2016; Mittal et al. 2018).

Figure 1 shows the topography for the SE dynamical core ((a), Topography-I) and, for comparison, the FV dynamical core ((b), Topography-II). It can be clearly seen that there are considerable differences between the height of the mountains with different smoothing operators and smoothing strengths (shown in Fig. 1(c)).

Therefore, manipulating the topography in each core can be a source of uncertainty, which introduces an additional source of error in the interpretation. To deal with this issue carefully, we devised sensitivity experiments using different topography sets on top of the dynamical core differences. The first is the 'SE(Topo-I) model', which uses the SE dynamical core on a cubed-sphere grid system with a generic topography (Topo-I) from the SE dynamical core distribution. Second is


the 'FV(Topo-I) model', based on the FV dynamical core on a longitude-latitude grid system with Topo-I. The third is the 'FV(Topo-II) model', which differs from the FV(Topo-I) model in only the generic topography used for the default FV dynamical core distribution (Topography-II). Details concerning the different prediction models with their dynamical core options are contained in here, while brief descriptions of the specific settings of each model are provided in Table 1 and 2.

### 2.1.3. Initial and Boundary Conditions

The initial conditions for the integration were prepared by using the Japanese 55-year reanalysis data (JRA-55) (Ebita et al. 2011). We interpolated the reanalysis variables, including the surface temperature at 2m, winds, radiation, specific humidity, precipitation, evaporation, and other climate parameters, to the model horizontal and vertical grids for the

initialization of the corresponding model variables. A 15-member ensemble was prepared with lagged initial conditions at 6-hour intervals up to the starting date (i.e., 1st, October, and 1st, December). The sea surface temperature (SST) and sea ice concentration (SIC) forecasts were used as global boundary conditions from the reforecast of the National Centers for Environmental Prediction (NCEP) Climate Forecast System (CFS; available from 1979 to 2010) (Saha et al. 2010) and CFS version 2 (CFSv2; available from 2011 to present) (Saha et al. 2014). The use of NCEP CFS/CFSv2 as a boundary condition

is based on the study of Lindsay et al. (2014), which reported that CFS/CFSv2 has less bias than the NCEP/National Center for Atmospheric Research (NCAR) reanalysis.

### 2.1.4 Model Hindcast Experiment

The SE(Topo-I), FV(Topo-I), and FV(Topo-II) models performed sub-seasonal predictions for boreal winter surface

air temperature. To evaluate the prediction performance, we conducted hindcast experiments for 17 winters from 2001/2002 to 2017/2018. For each year, the three months from October 1st are predicted, and the last two months (i.e., November to December) are averaged to define early winter predictions. For late winter predictions, ideally, we started from December 1st and used the last two months (i.e., January and February) for the calculation. These hindcasts provided us 15-ensemble members (1-day time-lagged), from which we used the ensemble mean when analyzing the model output.

### 2.2 Validation Data and Method

### 2.2.1 Validation Data

We used monthly mean European Centre for Medium-Range Weather Forecast (ECMWF) interim reanalysis (ERA-Interim) products for the prediction model validation (Dee et al. 2011). Data that were used to validate the predicted

temperature, winds, other climate parameters were provided on a 2.5º by 2.5 º latitude-longitude grid. We used the above data from November 2001 to February 2018, when the performance of the sub-seasonal prediction model was available. The validation and prediction data were constructed under the same sub-seasonal timescales, the early winter and late winter, from 2001/2002 to 2017/2018. Moreover, the anomalies were calculated by removing each season's climatological mean (from the





period between 2001/2002 to 2017/2018). The prediction data was interpolated at the same resolution as the ERA-Interim reanalysis products.

### 2.2.2 Validation Method

The skill score test for predictability is calculated using the ensemble mean instead of averaging individual ensemble members in all predictions. In this study, the prediction skill score is mainly calculated for boreal winter surface air temperature in climatological mean, followed by two important statistical skill score tests.

    We used two statistical skill score techniques applied to verify deterministic prediction skills, first is the simple anomaly correlation coefficient (ACC) and second is the mean-square skill score (MSSS) (Goddard et al. 2013; Choi et al.

2016). To calculate the ACC and MSSS, we use the following equations:

$$ACC(\tau) = \frac{\frac{1}{n}\sum_{j=1}^{n}[F_{j\tau}-\bar{F}_\tau][O_{j\tau}-\bar{O}_\tau]}{\sqrt{\frac{1}{n}\sum_{j=1}^{n}[F_{j\tau}-\bar{F}_\tau]^2}\sqrt{\frac{1}{n}\sum_{j=1}^{n}[O_{j\tau}-\bar{O}_\tau]^2}},$$
(1)

The ACC calculated the correlation coefficients between a set of forecast runs $F$ and observation $O$, where $O$ corresponds to reanalysis products in this study. Forecast ensembles can be characterized by $F_{j\tau}$, where $F$ is ensemble mean prediction in each

model's, $j$ is the initialization year, $n$ is the total number of experiments, and $\tau$ is the forecast lead time. The climatological averages of reanalysis products (observations) and forecast runs are calculated by the following formula:

$$\bar{O}_\tau = \frac{1}{n}\sum_{j=1}^{n}O_{j\tau} \text{ and } \bar{F}_\tau = \frac{1}{n}\sum_{j=1}^{n}F_{j\tau},$$
(2)

The MSSS is based on the mean-squared error (MSE) (Murphy. 1988), which is calculated by

$$MSSS(\tau) = \frac{\frac{1}{n}\sum_{j=1}^{n}[(F_{j\tau}-\bar{F}_\tau)-(O_{j\tau}-\bar{O}_\tau)]^2}{\frac{1}{n}\sum_{j=1}^{n}(O_{j\tau}-\bar{O}_\tau)^2},$$
(3)

The ACC calculates the linear association between an observation and forecast runs in models, while the MSSS calculates the

bias of prediction model as the relative magnitude. The MSSS is no statistically significant compared to the ACC, one may determine that the direction of anomalies is predicted well, but its magnitude is uncertain. Throughout the literature, subjective threshold values are conventionally used for evaluating statistical skill score results (i.e., ACC greater than about 0.5 and MSSS greater than 0).

    To evaluate the teleconnection linkage in boreal winter in the sub-seasonal prediction models, we used the Arctic

temperature (ART) indices proposed by Kug et al. (2015), which has an important interpretation for the linkage between Arctic and mid-latitudes, extending to the implication of the WACC patterns. The calculation method is quantified by the correlation





coefficients between ART1 index (T2M area-averaged over the Kara-Barents Seas; 70–80ºN, 30–70ºE) with the spatial T2M of East Asia (35-50ºN, 80-130ºE), and ART2 index (T2M area-averaged over the Chukchi-East Siberian Seas; 65–90ºN, 160–200ºE) and the spatial T2M of North America (35–50ºN, 230–280ºE). According to previous research, in recent decades, the ART indices show a trend towards a warmer the Arctic and colder mid-latitudes.

30



## 3. Results

### 3.1 Mean Surface Air Temperature at 2m (T2M)

We first examine the spatial patterns in seasonal climatology for different dynamical cores. In this study, the model climatology is evaluated using each seasonal prediction's ensemble mean and target observation (reanalysis products) for a hindcast period (2001-2018). We show the bias of the prediction model for the T2M mean state (Fig. 2). The 17-year climatology predictions for boreal winter T2M in each prediction model are compared with the ERA-Interim reanalysis product.

Overall, the mean T2M among prediction models generally reproduce the observed climatology over the oceans well. However, over the land surface, obvious bias patterns are observed among prediction models in both seasons, which are shown in Fig. 2(b)–(d) (early winter), and 2(h)–(j) (late winter). Prediction models showed similar bias patterns over the northern hemisphere in both seasons. Particularly, a cold bias is found in the Arctic and part of North America and East Asia, whereas a warm bias is found in broad areas over Greenland and Europe compared to the ERA-Interim. The SE(Topo-I) model predicts a warmer temperature over the Northern Hemisphere than FV models (shown in Fig. 2(e)–(f) and (k)–(l)). This implies that the SE dynamical core tends to predict warmer temperatures than the FV dynamical cores during both seasons.

Figure 2 (e) and (k) show the difference in mean T2M bias between the SE(Topo-I) and FV(Topo-I) in the early and late winter, respectively. With the same topography, the SE(Topo-I) model predicts warmer temperatures (about 1–2 ℃ in mean T2M) than FV(Topo-I) in early winter. Similarly, the late winter shows the same warm bias pattern between the two models, but not in the Arctic. With only the topographic differences in the grid formation, the warmer bias is found in the SE(Topo-I) and FV(Topo-II) during both seasons (about 3–4 ℃ in mean T2M) (shown in Fig. 2(f) and (l)). It shows a significant difference compared to the differences in FV(Topo-I) models, especially the obvious differences found in Eurasia that are related to elevation from the topography (e.g., mountain area). In the mean T2M state, the SE dynamical core consistently shows a warm T2M bias pattern, particularly in North America and Greenland. Even without the effect of the topography from the grid formation, the SE dynamical core can show a tendency to predict warmer temperatures than the FV dynamical core. However, if differences in dynamical cores are not given at the same topography effect, the bias is even more significant.

### 3.2 Forecast Skill

To examine the prediction skills in sub-seasonal time scale, the ACC between prediction's anomalies and reanalysis products are calculated for the ensemble mean determined from 17 winter seasons, and Figure 3 shows the ACC of the T2M anomaly for the prediction models in both seasons. The black dots show statistically significant ACC regions (at the 95% confidence level). Since the forecast skill from both dynamical cores in both seasons over the ocean is almost identical (shown in Supplementary Fig. 1), we masked out the ocean for clarity and analyzed the differences over the land area before describing the changes in forecast skill ascribed to dynamical core differences.

In the early winter, prediction models show a positive ACC broadly over Eurasia, although there exists little significant signal over the area. The FV(Topo-II) model shows a negative signal over Eurasia, but a significantly positive ACC



in East Asia. Overall, present models commonly show low prediction skills in North America. In the Arctic region, on the other hand, these models show high prediction skills, particularly in the Kara-Barents Sea and the Chukchi Sea, where the variability in sea ice is greatest (shown in Supplementary Fig. 1). The SE(Topo-I) model, in particular, shows a significantly higher ACC in a broad area around the Arctic compared to the FV(Topo-I) model, due to differences in the dynamical cores

despite having the same topography. In late winter, the ACC decreased over the Arctic region relative to early winter. Overall, the prediction skills of North America increased relative to early winter; mainly, the SE(Topo-I) model shows a highly positive ACC over North America. The FV models also show an improved prediction skill over East Asia and Eurasia.

Similar to Fig. 3, Fig. 4 shows the spatial distribution of the mean-squared skill score (MSSS) for winter T2M (also the MSSS masked out the ocean for the same reasons as Fig. 3, shown in Supplementary Fig. 2). MSSS score is meaningful

only when greater than 0. If the MSSS skills increase (red color), it means that the prediction would yield the best performance with a certain prediction model. In both seasons, the MSSS score is consistent with the ACC patterns among prediction models. The blue colored area has low predictive skills in sub-seasonal timescales. In early winter, same as the ACC, the SE(Topo-I) model has a higher predictive ability around the Arctic region than the FV models. Moreover, the FV(Topo-II) model has a positive MSSS score in East Asia. In late winter, the SE(Topo-I), FV(Topo-I), and FV(Topo-II) models show improved

predictive skill scores over North America, East Asia, and Eurasia, respectively, relative to early winter, which is consistent with Fig. 4. Subsequently, the SE(Topo-I) model has better prediction skills in North America in late winter, and FV models have higher predictive performances in East Asia and Eurasia in late winter. It implies that prediction models show different predictive skills depending on the dynamical cores on the grid formation, particularly over the Arctic region and the land surface of the mid-latitudes.

### 3.3 Zonal Mean Vertical Distribution

Figure 5 shows the zonal mean vertical distribution using climate parameters, comparing between the SE(Topo-I) and FV(Topo-I) models during the early winter and late winter. The contour line denotes the FV(Topo-I), and shading represents changes produced when using SE(Topo-I) (SE minus FV). We show the seasonal climate parameters in the averaged zonal

mean fields using the temperature, transient eddy momentum flux, transient eddy heat flux, and vertical velocity because these seasonal climate parameters control many significant features of global climate, including the distribution of pressure and temperature, and the meridional transport of heat flux.

Compared to the FV(Topo-I) model, the SE(Topo-I) predicted a warmer vertical temperature distribution into the stratosphere near the 60ºN region in early winter (Fig. 5 (a)) and a colder stratosphere over the subtropics in the late winter

(Fig. 5 (e)). Not shown in Fig. 5, the SE(Topo-I) compared to the FV(Topo-II) also showed a similar zonal mean vertical distribution as the FV(Topo-I); but the differences are much higher than compared to the FV(Topo-I). Notably, in early winter, warmer temperatures (> ~1 ºC) are observed in the Arctic region when using the SE(Topo-I) model, compared to the FV(Topo-II), the values are about 1.4 ºC. The characteristics of these zonal mean temperature differences are consistent with the preliminary result from using the SE dynamical core in the Arctic region. Further, we examine the possible contribution of



differences between the SE(Topo-I) and FV(Topo-I) in the transient eddy momentum and heat fluxes, and the vertical velocities during both seasons. Fig. 5 (b) and (f) display the difference in the transient eddy momentum flux, wherein the weakest momentum flux is 30ºN near 300 hPa in early winter, and the increased momentum flux is in the tropics above tropopause in late winter. It is associated with a colder temperature signal in late winter (shown in Fig. 5 (e)) due to increasing

weaker momentum flux, decreasing momentum flux convergence, and weakening in the mean meridional circulations over the subtropics. The SE(Topo-I) model also predicted a strengthen transient eddy heat flux at the surface to the center of 60ºN in the tropopause. Also, it shows that a weakening in the transient eddy heat flux during early winter over the Arctic region, relative to the FV(Topo-I) model. It means that the SE(Topo-I) model tends to shift to the southward in the maximum transient eddy heat flux. Fig. 5 (g) shows that the SE(Topo-I) model tends to heat flux weakens over the subtropics in the late winter,

with the transient eddy momentum flux. It indicates that the SE(Topo-I) model simulated to changing fluctuations in mean circulation patterns, making a rising and sinking motion over the subtropics, and the Arctic region's tropopause, respectively. It caused by the indirect effects from the forcing circulation patterns in both seasons (shown in Fig. 5 (d) and (h)).

As a result, the SE-Model predicted a strengthening sinking motion in the Arctic region at the tropopause through adiabatic warming, enforcing a warmer vertical temperature distribution. These warming patterns in the Arctic region also

manipulated by increasing the transport of poleward eddy heat flux over the mid-latitudes. As the predictive skills, the warmer Arctic vertical temperature in the SE(Topo-I) compared to the FV(Topo-I) is associated with the prediction of mid-latitudes due to the differences in mean circulation patterns despite using the same physics. It showed that, despite using the same topography, depending on the characteristics of the dynamical cores, the SE(Topo-I) model showed warmer vertical temperature distributions over the Arctic region compared to the FV(Topo-I) and FV(Topo-II) also. It showed similar results

for both FV(Topo-I) and FV(Topo-II), but the difference is greater when compared to the FV(Topo-II). Ultimately, the distribution of the vertical temperatures in the SE(Topo-I) model has enhanced sinking motion effects, resulting in a difference in mean circulation patterns.

### 3.4 Model Representation of the Arctic-Midlatitude Linkage

Recent studies have demonstrated that a warmer Arctic is contributing to colder winters across the Northern Hemisphere continents (Overland and Wang 2010; Serreze and Barry 2011). A warmer Arctic climate is predicted for the future in many studies (Pithan and Mauritsen. 2014; Screen 2017; Jang et al. 2019). However, the prediction skills of a warm Arctic-mid-latitude linkage that can have the potential to amplify extreme mid-latitude weather events are still undergoing. Particularly, the Kara-Barents Sea to East Asia and the Chukchi-East Siberian Sea to North America are of significant interest

for Arctic-mid-latitude linkage impacting the predictive skill of extended-range weather forecasts (Jung et al. 2015).

We examine how well prediction models with different dynamical cores predict the winter Arctic-mid-latitude linkage using the Arctic temperature (ART) indices with a spatial distribution around the Northern Hemisphere. We used two ART indices defined by Kug et al. (2015) (detailed calculation methods in Chapter 2). Figure 6 shows a correlation between ART indices and T2M for the early winter and late winter. Figure 6 (c)–(f) show the correlation patterns with the ART1 (Kara-



Barents Seas) and Figure 6 (g)–(l) show the correlation with the ART2 (Chukchi-East Siberian Seas), with the SE(Topo-I) and FV(Topo-I) models in both seasons. The time-series of ART indices in the SE(Topo-I) and FV(Topo-I) models capture the year-to-year Arctic temperature variability during the wintertime (shown in Fig. 6 (a)–(b)). Both indices exhibit strong positive correlations over their region (green box) for both models (correlation coefficients ≈ 0.85) but show different correlation

5    patterns in much of the mid-latitudes.

For ART1 (shown in Fig. 6 (c)–(f)), SE(Topo-I) and FV(Topo-I) show a negative correlation prevailing over Eurasia in early winter. The FV(Topo-I) model found more negative correlations in this area than SE(Topo-I). Although both models show no relationships over Eurasia in late winter, both models show a strong negative correlation in East Asia (including South Korea and Japan) in late winter. For ART2 (shown in Fig. 6 (g)–(j)), the SE(Topo-I) and FV(Topo-I) models show the weakest

10   relationship between ART2 and North America in early winter, despite both models capturing the ART2 areas well. In late winter, the SE(Topo-I) shows a negative correlation in North America, while the FV(Topo-I) has a positive correlation in this area. This is consistent with the T2M prediction skills for the SE(Topo-I) and FV(Topo-I) models during both seasons. These results imply that the SE(Topo-I) captures warmer Arctic and colder North American temperatures relatively well, while FV(Topo-I) model captures warmer Arctic and colder Eurasian temperatures, alternatively.

30





## 4. Discussion and Summary

This study has examined the sub-seasonal prediction skills for boreal winter temperatures using two different dynamical cores. The predictive skill differences between the two dynamical cores are evaluated using hindcast runs for the period 2001/2002 to 2017/2018 and compared with the reanalysis product (ERA-Interim). Compared to the FV dynamical core, in general, the SE dynamical core predicts a warmer Northern Hemisphere in both winter seasons. For ACC/MSSS skill score tests, the FV dynamical core has a significant correlation coefficient over East Asia in early winter. In contrast, the SE dynamical core has improved the correlation coefficient over North America in the late winter. We summarize the differences in prediction skill for T2M for SE(Topo-I) (red box), FV(Topo-I) (blue box), and FV(Topo-II) (black box) in the area-averaged Northern Hemisphere, Arctic region, East Asia, and North America, respectively (Fig. 7). In Fig. 7, the domain definition for the box averaging for each region follows Kug et al. (2015). From the dynamical analysis of zonal mean differences between SE(Topo-I) and FV(Topo-I), we suggest that the warmer Arctic temperatures in the SE(Topo-I) model compared to the FV(Topo-I) is due to the difference in mean circulation patterns. Differences in circulation patterns are further related to the enhanced adiabatic warming effect caused by the distribution of vertical temperatures in the Arctic region, which are characteristics of the dynamical core (not figured, but the differences in FV(Topo-II) showed similar results). Moreover, the temperature teleconnection patterns between the Arctic and mid-latitudes in both prediction systems show a significant relationship with the ART Index. For ART1, the FV(Topo-I) has great predictive skills in East Asia in early winter, and the SE(Topo-I) has a great predictive skill in North America with the ART2 index in late winter.

Thus, this study investigates the effect that choosing a particular dynamical core has on the predictability of sub-seasonal atmospheric prediction models. We find that the predictive skills of these two dynamical cores are significantly different despite the almost identical physical parameterization used in models. By this, we raise concerns about the choice of dynamical cores in the sub-seasonal and seasonal predictability studies, especially for the simulation of Arctic climate and related teleconnection patterns in those time-scales. We believe our study provides an initial motivation for more studies regarding the optimal choice of dynamical cores in the future development of the climate prediction models.



**Author Contributions**

Ha-Rim Kim conducted the analyses and took the lead in writing the manuscript. Baek-Min Kim and Sang-Yoon Jun developed the idea of this research. Yong-Sang Choi provided feedback and helped shaped the research. All authors discussed the results and contributed to the final manuscript.

**Competing Interests**

The authors declare no competing interest.

**Acknowledgements**

This research was supported by the research project of Korea Polar Research Institute (KOPRI) titled 'Predictability study on the Arctic-midlatitude teleconnection using polar oceanic and cryospheric surface observations' (PE20090) and the Basic Science Research Program through the National Research Foundation of Korea (NRF)' funded by the Ministry of Education (2018R1A6A1A08025520).

**Code Availability**

The source code in this study is based on the National Center for Atmospheric Research/University Corporation for Atmospheric Research (NCAR/UCAR) Community Earth System Model (CESM) version 1.2.1 at revision 74732 whose code can be acquired from the CESM1 SVN repository (https://svn-ccsm-models.cgd.ucar.edu/cesm1/release_tag/cesm1_2_1). We set up two dynamical cores (finite volume core and spectral element core) with the same CAM4 physics by using configuring

script of the atmospheric model CAM5 included in this CESM version. Flag sets we used in the standard configuration scripts for each dynamical core as summarized in the table below.

**Summary of flag sets for the two dynamical cores used in this study:**

|  | FV (finite volume) dynamical core | SE (spectral element) dynamical core |
|---|---|---|
| **Dynamical core, horizontal resolution, and physics settings in CAM5 configuration (when using *configure* script)** | -dyn fv -hgrid 1.9x2.5 -phys cam4 -ocn docn | -dyn se -hgrid ne16np4 -phys cam4 -ocn docn |
| **Related case and resolution settings in the CESM1 configuration (when using *create_newcase* script)** | -case F -res f19_g16 | -case F -res ne16_g16 |





## Data Availability

All data that we used are freely and publicly available as follow links: JRA 55-year Reanalysis (https://jra.kishou.go.jp/JRA-55/index_en.html#download), NCEP CFS/CFSv2 (https://www.ncdc.noaa.gov/data-access/model-data/model-datasets), ECMWF ERA-Interim Reanalysis (https://apps.ecmwf.int/datasets/data/interim-full-moda/levtype=pl/).

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





**Table List**

Table 1. Reference for information of sub-seasonal prediction models referred to within the text.

| Model name | Dynamical core | Resolution | Topography | Physics package | Prediction years |
|---|---|---|---|---|---|
| SE(Topo-I) | Spectral element | Cubed-sphere, approx. 2.0º (named as ne16np4) | Topography-I (topography was set to match the SE dynamical core) | * CAM5 (with CAM4 physics including coefficients from the dynamical core's default setting) | 2001/2002 – 2017/2018 |
| FV(Topo-I) | Finite volume | 1.9º Lat x 2.5º Lon (named as fv19) | | | |
| FV(Topo-II) | Finite volume | 1.9º Lat x 2.5º Lon (named as fv19) | Topography-II (topography from the default finite volume dynamical core) | | |

* Listing includes the dynamical core, horizontal spatial resolution, and version of CAM physics used. CAM5 with CAM4 physics setting refers to the CAM4 physics package using the parameter settings that provided coefficients for the default dynamical core (see section on the code availability).

Table 2. Reference for information of detailed physics schemes in the sub-seasonal prediction models

| Model Physics Package | | |
|---|---|---|
| Model Descriptions | Model System | CAM5 Atmospheric model system |
| | Ensemble Generation Method | 1-day time lagged |
| | Ensembles | 15 |
| | Resolutions | 1.9º Latitude x 2.5º Longitude |
| | Levels | 30 |
| Initial Conditions | Atmosphere (ATM) | JRA-55 Reanalysis |
| | Land (LND) | Long-term spin-up run (30-years using the climatology) |
| Boundary Conditions | Sea Surface Temperature (SST) | NCEP-CFS/CFSv2 Forecast SST and SIC |
| | Sea Ice Concentration (SIC) | |



**Figure Lists**

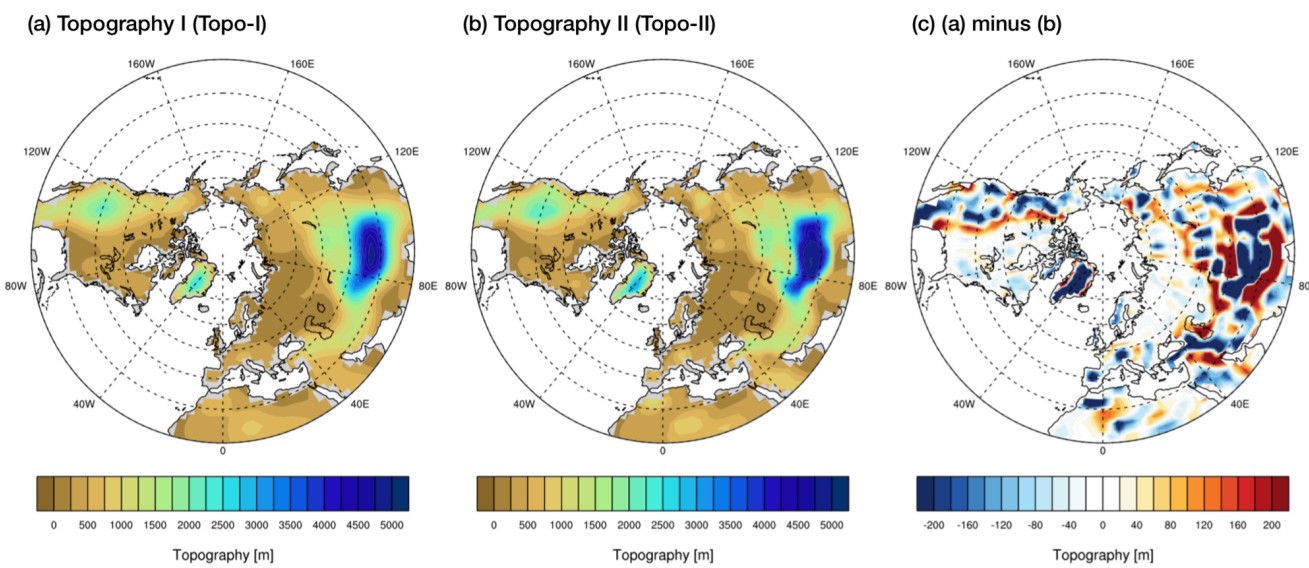

Figure 1: Spatial distribution of topography for spectral elements (SE) dynamical core and finite volume (FV) dynamical core. (a) Topography I (Topo-I) from the SE dynamical core, (b) Topography II (Topo-II) from the FV dynamical core, and (c) difference between (a) and (b).



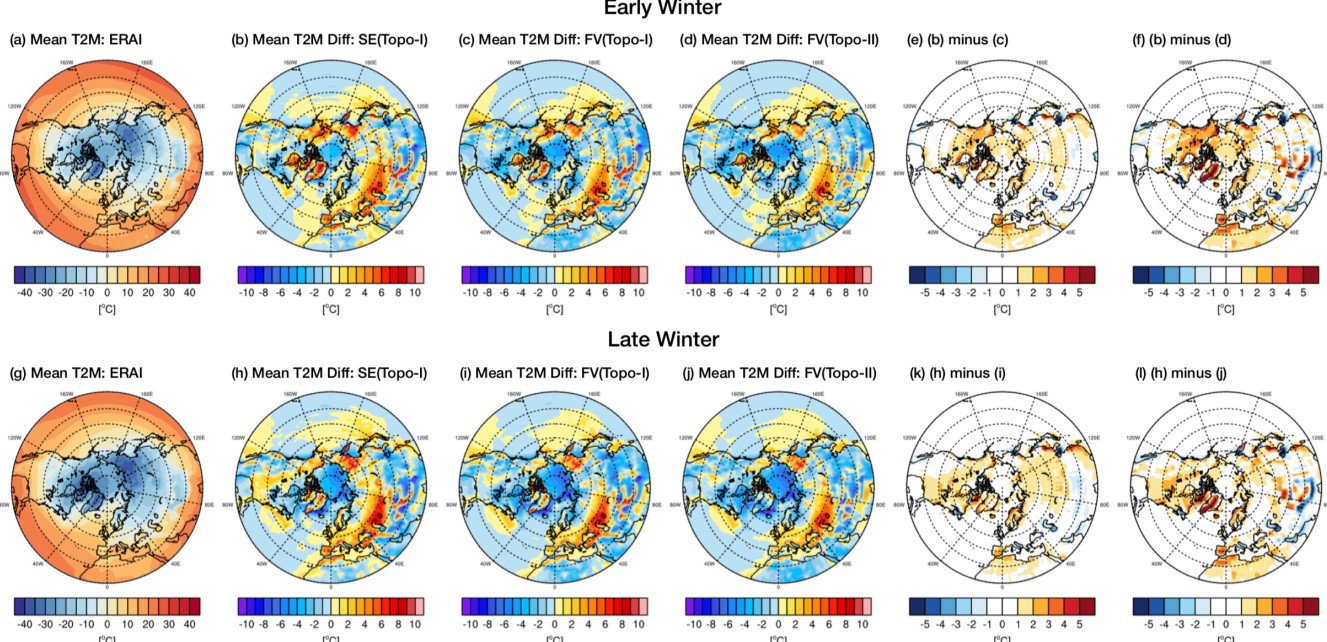

Figure 2: Spatial distribution for mean 2m-air temperature (T2M) for the early winter and late winter (2001/2002 – 2017/2018) in two sub-seasonal prediction model; (a) mean T2M from ERA-Interim reanalysis (ERAI), (b) mean T2M model bias of SE(Topo-I), (c) mean T2M model bias of FV(Topo-I) (d) mean T2M model bias of FV(Topo-II), and (e) – (f) mean T2M difference between SE(Topo-I) and FV models. Same with (g)-(l) except for late winter.

Figure 3: Spatial distribution of anomaly correlation coefficients (ACC) for 2m-air temperature (T2M) between models and ERA-Interim reanalysis (ERAI) during the early winter and late winter (2001/2002 – 2017/2018); (a) ACC between SE(Topo-I) and ERAI, (b) ACC between FV(Topo-I) and ERAI, (c) ACC between FV(Topo-II) and ERAI. Same with (d)-(f) except for late winter. The black dots indicate statistical significance at the 95% confidence level. A mask has been applied such that only ocean grid points for a clear distinction of ACC with the land area.

Figure 4: Spatial distribution of mean-squared skill score (MSSS) for 2m-air temperature (T2M) between models and ERA-Interim reanalysis (ERAI) during the early winter and late winter (2001/2002 – 2017/2018); (a) MSSS between SE(Topo-I) and ERAI, (b) MSSS between FV(Topo-I) and ERAI, (c) MSSS between FV(Topo-II) and ERAI. Same with (d)-(f) except for late winter. (MSSS showing greater than -1.0). A mask has been applied such that only ocean grid points for a clear distinction of ACC with the land area.

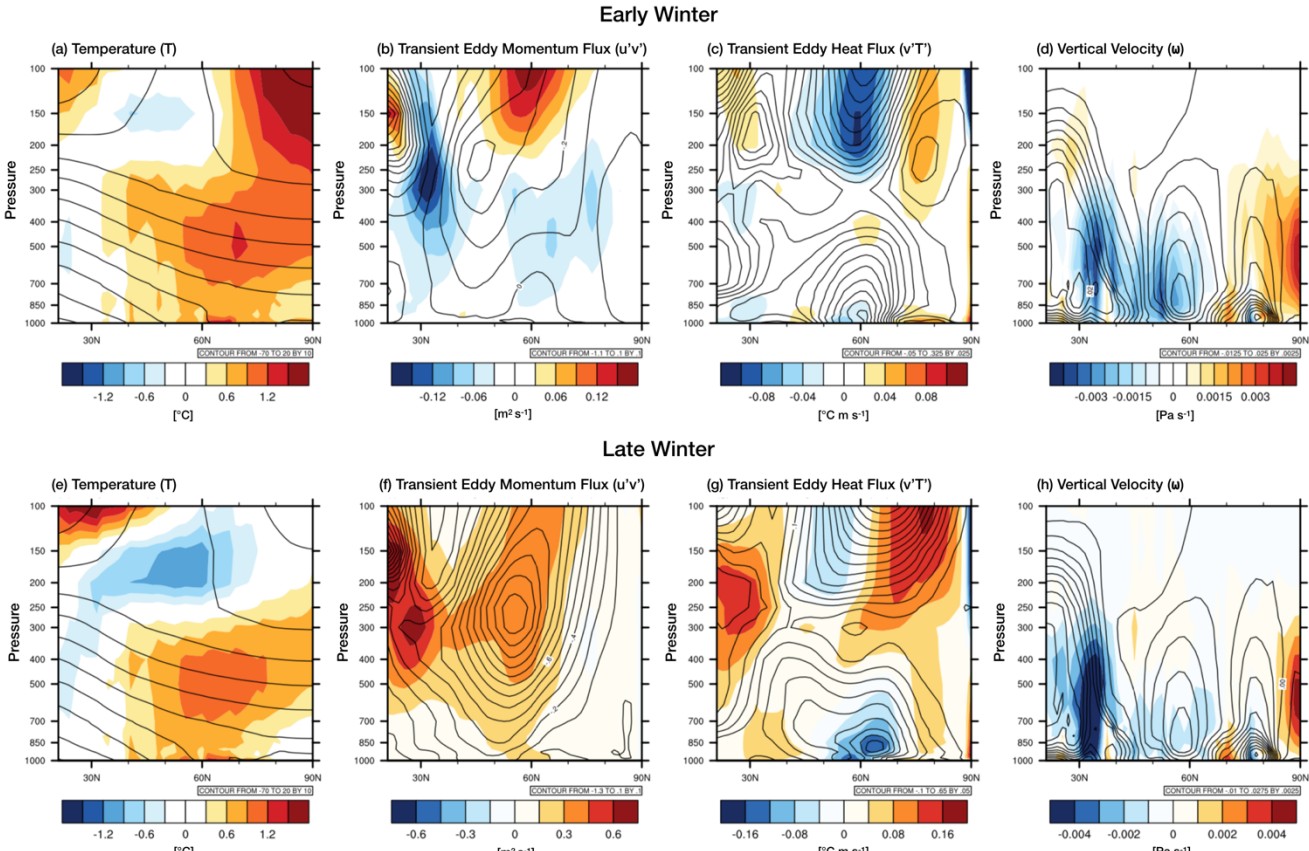

Figure 5: Zonally averaged climate parameters during the early winter and late winter (2001/2002 – 2017/2018) produced by SE(Topo-I) and FV(Topo-I); the contour indicates the FV(Topo-I), and shading indicates a difference between SE(Topo-I) and FV(Topo-I); (a) temperature, (b) transient eddy momentum flux, (c) transient eddy flux, and (d) vertical velocity. Same with (e)-(h) except for late winter.





Figure 6: Correlation coefficients of 2m-air temperature (T2M) anomalies for SE(Topo-I) and FV(Topo-I) with respect to ART1 and ART2 (ART: Arctic temperature) indices during the early winter and late winter (2001/2002 – 2017/2018); the green box indicates the each Arctic region of ART indices (ART1: Barents-Kara Seas, ART2: Chukchi-East Siberian Seas); (a) – (b) T2M time-series of the ART1 and ART2 for the SE(Topo-I) and FV(Topo-I) in ealy winter and late winter, respectively. (c)-(d) correlation relationship for T2M with ART1 in early winter, and (g)-(h) for late winter; (e)-(f) correlation relationship for T2M with ART2 in early winter, and (i) –(j) for late winter. The black dots indicate statistical significance at the 95% confidence level.





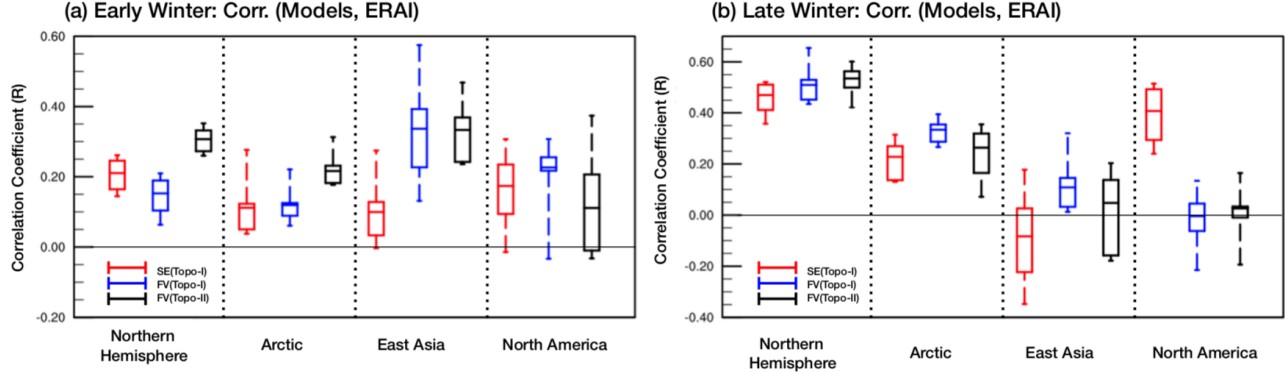

Figure 7: Box plots of the correlation coefficients between models and ERA-Interim reanalysis (ERAI) for each mean ensemble (ensemble number is 1 to 15) of SE (Topo-I) (red box), FV(Topo-I) (blue box), and FV(Topo-II) during the (a) early winter and (b) late winter; median lines indicates the mean values from each mean ensemble of models; x-axis denotes the area-averaged region (Northern Hemisphere, Arctic, East Asia, and North America).