# Peer review of "Altered sub-seasonal predictability of Community Atmosphere Model 5 (CAM5) in CESM 1.2.1 by the choices of dynamical core"

_Geoscientific Model Development, 2020_

## Referee Comment (RC1) · Anonymous Referee #1 · 27 May 2020

General comments

The authors used FV and SE version of CESM and evaluated how do change in dynamical core alternate the skill score of T2m in winter at a S2S time scale and try explaining why the difference happens. However, researches have already showed that forecast results depend on choice of dynamical core as reviewed by the authors (e.g., Jun et al. 2018) and ECMWF's idealized experiments (https://doi.org/10.5194/gmd-12-651-2019). I understand it is model evaluation paper so the authors would not show how the dycores are different and how the difference alternate forecast. Even in considering this point, the model evaluation is not enough (e.g., just looked at T2m). I feel it

is questionable if the manuscript is valuable for weather/climate modeling community as well as the S2S research community. Also, presentation of the results is poor and sometimes I could not understand what the authors intended to say. Thus, I would like to say the manuscript is not suitable for publication in GMD and put some comments for improving the manuscript.

Specific comments

P2 L24 I could not find new from Jun et al (2018) that performed comprehensive model evaluation using various climate models.

P4 Section 2.1.1 The authors mentioned difference in horizontal grid. How was vertical grid? Also I suppose that changes in dycore cause difference in numerical stability. How is the numerical diffusion in the both dycores?

P4 L24-26 I understood "strong high-order interpolation improves the quality of the topography" However, obviously, the topography is smoother than that in FV. I was confused about that.

P4 L27 Fig 1 Because horizontal grid is different between FV and SE, you need to regrid the topography to calculate the differences. In general, the regridded topography highly depends on which method you used. In addition, how is the difference over the ocean? I guess SE has some topography over the ocean.

P5 L8-9 Are the diagnostic variables (e.g., the T2m, precipitation, evaporation) really necessary for the model initialization?

P7 L3-4 "previous research" which one?

P8 L24-25 "However..." I could not understand what you intended to say.

Section 3.2 In general, the region pointed is not clear and exaggerated. For example, The authors state "a positive ACC broadly over Eurasia (P8 L33)", but it is not true for the eastern part. So I would just say "a significant positive ACC over Europe" and delete

next sentence. In addition to that most part of the section can not be understood without supplemental figure 1. For example, the authors mentioned about Kara-Barents Sea and the Chuchi Sea in P9 L2 but data is not shown in Fig 2. Why not to replace Fig 2 with Fig S1. Indicating these two sea in the figure is helpful for readers.

Section3.3 Suddenly the authors start to focus on difference between SE and FV with Topo I. But I am wondering why the authors did not show comparison with the observation. Also I could not understand many part of the section. I need more kind explanation.

P9 L34 what is "preliminary result?"

P10 L4-6 "It is associated…over the subtropics" I could not understand this long sentence. I need more kind explanation.

P10 L7 "it shows … Arctic region" I could not understand this sentence. I need more kind explanation and where is "Arctic region?"

P10 L8-9 "It means…eddy heat flux" I suppose you are talking about Fig 5(g). I found there is tripole structure. So we could not say which direction it is shifted.

P 10 L9-11 "Fig 5(g)…respectively." I could not understand this sentence. I need more kind explanation.

P10 L33 "Fig 6 shows a correlation" It is not true for Fig6a and 6b. Why not to include the observation to Fig 6a and 6b?

P11 L7 I suppose SE should be FV and FV should be SE. but I am not sure...

P11 L11 "a negative correlation" Where?

---

## Referee Comment (RC2) · Juan Antonio Añel (Referee) · 19 Oct 2020

In this work, the authors intend to compare the impact on sub-seasonal forecast derived from the use of two different dynamical cores for CESM. This is what they claim; however, the development of the manuscript diverges from this goal. There are several main problems with the current version of this work.

- Lack of clarity in the storyline along the manuscript: The authors mix in the discussion very different issues, without to reach a clear conclusion on the impact of the dynamical core on the different results of predictability. And, as they acknowledge in the text, they are great. These issues involve the dynamical cores, the role of the orography or

uncertainties in the Arctic region. In the last part of the manuscript, the discussion on the Arctic gets a lot of focus. This is done at the expense of the discussion of the impact on sub-seasonal predictability.

- Lack of interpretation of the results: Mostly, the manuscript presents the results without a profound analysis of the potential reasons for them.

- Too many uncertainties involved and not well explained: The authors acknowledge the need to use different topographic schemes because of the different cores. To overcome this problem, they perform several simulations with diverse topography. However, its impact on the results obtained is barely discussed.

- In the manuscript, only two fields are analysed: surface temperature and the eddy momentum flux. In work on sub-seasonal predictability for the boreal hemisphere, I would expect that the analysis of variables and fields was more complete. As an example, patterns such as the Arctic Oscillation play a fundamental role, and nothing is said in the manuscript about it.
* * *

---

## Editor Comment (EC1) · Juan Antonio Añel (Editor) · 19 Oct 2020

Dear authors,

After reading your manuscript, both one reviewer and I consider that it must be rejected in its current form. You can read my comments as one of the referees. One of the issues is that I have struggled to secure reviewers for your work, despite having tried a long list of potential ones. This has delayed the decision on your submitted work much more than it is desirable and I apologize for it. I consider that the storyline of your work needs to be completely reshaped. Also, you have to work on the attribution of the results, present a full analysis, including other 'variables' and clarify the message.

Best regards,

Juan A. Añel Geosc. Mod. Dev. Topical Editor